# HOW NEURAL NETWORKS WITH DERIVATIVE LABELS WORK: A NEURAL TANGENT KERNEL PERSPECTIVE

## ABSTRACT

Deep neural networks have achieved impressive results in a range of fields, while their analytical properties have been slow to develop. Recently, a theoretical tool called Neural Tangent Kernel (NTK) has been proposed and extended to diverse architectures based on neural networks. This tool helps explain their convergence and generalization with a least-square loss. However, researchers in numerous fields have trained their networks using an additional derivative loss item, such as Jacobian Regularization and PINN (Physics-Informed Neural Networks). This loss setup has generality, while it often leads to challenging convergence issues. To explain this phenomenon, we propose a general paradigm that utilizes Gâteaux derivative labels to describe all these tasks by extending tools of NTK. We provide two kinds of perspective, inductive perspective of convergence equations and geometrical perspective of parameter updating directions, to explain the hard convergence. We also conduct experiments to verify our propositions. Finally, we provide specific expressions for distinct tasks within our paradigm.

## 1 INTRODUCTION

Deep neural networks with a fully-connected network (FNN) architecture have demonstrated remarkable performance across various fields. While the fundamental premise is that FNNs can effectively approximate any function given a sufficient number of neurons Hornik et al. (1989), the theoretical understanding of how these networks converge towards their objectives remains somewhat elusive. A promising avenue for addressing this is the exploration of infinite-wide networks, underpinned by certain invariants of expectation derived from initialization distributions, such as Mean Field Theory (MFT) Poole et al. (2016) Karakida et al. (2019) and Neural Tangent Kernel (NTK) Golikov et al. (2022) and Neural Network Gaussian Process(NNGP)Lee et al. (2017).

In recent years, there has been significant attention directed towards the Neural Tangent Kernel (NTK), a constant associated with infinite-wide networks in relation to their parameters. The NTK has successfully shed light on the convergence behavior of infinite-wide networks towards their objectives. However, its convergent equation is currently limited to the least-square loss, whereas multi-loss training is a widely employed approach in practical applications. The choice of loss function plays a pivotal role in the training process, typically involving the network's output and corresponding labels.

Recently, a novel form of loss formulation has gained attention, which includes an additional component derived from the derivative of the network's output. This additional component can be seen as a form of interpolation. Such tasks have gained prominence across various fieldsBubeck & Sellke (2021)Hoffman et al. (2019)Rodini (2022). These forms encompass various aspects, including regularization, embedded information, and modeling constraints.

For illustrating our topic, we firstly provide a succinct overview of the Gâteaux Derivative, which serves as a tool for characterizing the general input of tasks that treat samples as elements within a linear norm space. Define network as a operator in linear norm space $f : X \rightarrow R$, input such as images, coordinate can be described as such definition.

$Definition$ **1** Gâteaux Derivtative: An operator $f$ is a mapping from a linear norm space $X$ to real space $R$ Yosida (2012), Gâteaux Derivtative is an extension of directional derivative from real

space that is defined as:

$$\partial_h f_t(x) := \lim_{\epsilon \to 0} \frac{f_t(x + \epsilon h) - f_t(x)}{\epsilon} \tag{1}$$

in which $x, h \in X$ and $\epsilon, \partial_h f_t(x) \in R$. This definition can abstractly include the first-order loss term of input space and output of most tasks so that following analysis can be widely used in such tasks.

This article explores a specific type of multi-loss scenario, which we refer to as the "derivative loss". We delve into a generic derivative-based task involving datasets with perspective of interpolation: input samples denoted as $Tr \subset X$, their corresponding labels $\mathscr{L} : Tr \to R$, and their derivative labels $\mathscr{L}' : Tr \to R$. In theory, when employing a model with parameters equal in number to the training samples, zero error can be achieved. Similarly, as shown in Hornik et al. (1989), the same result can be attained when some of the data consists of input-label pairs while the rest comprises derivative labels related to the input. However, in the context of neural networks, such analysis becomes challenging due to the complexity of their derivatives Rodini (2022). Moreover, given their non-convex nature and the standard training strategy, Stochastic Gradient Descent (SGD), achieving this kind of optimality is not guaranteed.

Intuitively, extending the use of derivative labels $\mathscr{L}'$ imbues the network with additional capabilities for a given task. Yet, empirically, after introducing an extra loss component, predicting the network's convergence is not as straightforward as in interpolation. In this subsection, we present a paradigm for generic training with derivative labels based on the Neural Tangents Kernel (NTK). We expand the kernel to encompass two invariants for the derivative loss component and then derive the kernel for our paradigm through induction.

$$L(\boldsymbol{x}) \triangleq \frac{1}{2}\lambda_r \|f_t(\boldsymbol{x}) - \mathscr{L}(\boldsymbol{x})\|_2^2 + \frac{1}{2}\lambda_g \|\partial_h f_t(\boldsymbol{x}) - \mathscr{L}'(\boldsymbol{x})\|_2^2 \tag{2}$$

in which we set weights $\lambda_r$ and $\lambda_g$, where $\boldsymbol{x}$ represents all the training samples. This derivative label setting has been employed in various tasks (e.g. when $\mathscr{L}'(\boldsymbol{x}) = \boldsymbol{0}$, it's Jacobian RegularizationHoffman et al. (2019)), offering additional improvements, but it can also impact the convergence negatively. For instance, when using Jacobian Regularization in image tasks, the training results indicate that image classification maintains robust predictions even under certain perturbations. However, in such cases, the precision of image prediction without perturbations tends to decrease.

Other than that, we omit the definition of the network, including depth, rules of induction, and so on. For all our propositions, along with their proofs, we only include the definition of NTK itself. We refer readers to Jacot et al. (2018) for a rigorous and formalized introduction to NTK, or to Golikov et al. (2022) for a brief introduction. The NTK is introduced in our article using the gradient of the output of the randomly Gaussian-initialized network with respect to its parameters, as follows:

***Definition* 2** Neural Tangent Kernel (NTK): NTK is defined as (Omit the sign for the number of network layers):

$$\hat{\Theta}(x, x') \triangleq \nabla_\theta^T f_t(x) \nabla_\theta f_t(x') \overset{w \to \infty}{\Longrightarrow} \mathbb{E}_\theta \left[ < \nabla_\theta f_t(x), \nabla_\theta f_t(x') > \right] \tag{3}$$

as the width $w$ of network $f$ increases significantly, the kernel $\hat{\Theta}(x, x')$ approaches a constant value for $t$. This behavior depends solely on the structure of network $f$ and input $x$. It's worth noting that while the Neural Tangents Kernel (NTK) was initially proposed for Feedforward Neural Networks (FNNs), recent research Samarin et al. (2020), Arora et al. (2019a), Hron et al. (2020), and Franceschi et al. (2022) has extended Equation (2) to encompass diverse architectures, including CNNs, attention layers, and GANs. The key insight is that in the infinite-width limit, the convergence of a neural network trained by the least-squares loss function can be induced as follows:

$$\partial_t f_t(\boldsymbol{x}) = \hat{\Theta}(\boldsymbol{x}, \boldsymbol{x})(\boldsymbol{y} - f_t(\boldsymbol{x})) \tag{4}$$

which is a typical a system of first-order linear differential equations, taking NTK as its basic solution matrix, thus its solution is:

$$f_t(\boldsymbol{x}) = f_t(\boldsymbol{x}) - (I - e^{\hat{\Theta}(\boldsymbol{x}, \boldsymbol{x})t})(f_t(\boldsymbol{x}) - \boldsymbol{y}) \tag{5}$$

in which $\boldsymbol{x}$ represents vector of all training samples, $\boldsymbol{y}$ the correspondent labels, which means $f_t(\boldsymbol{x}, t)$ tends to converge to labels $\boldsymbol{y}$ with a large enough training step $t$, which is equivalent to kernel regression with kernel NTK.

We primarily focuses on examining the properties of networks trained with derivative labels (as defined in definition 1) using NTK tools (as defined in definition 2). We start by introducing a comprehensive definition of a general loss component that incorporates derivative labels. Following that, we provide an analysis of the NTK with the derivative loss version and offer a geometric perspective on both the regular and derivative components. In the latter part of the article, these insights are used to discuss batch properties. We further validate our theoretical findings with experimental results, approaching them from two distinct perspectives. Finally, we delve into a discussion of two forms of input for PINN, namely vectors and image adversarial Jacobian Regularization in matrix form and discuss batch setting in NNs with our proposition. Our contributions can be summarized as follows:

- Propose a comprehensive paradigm that provides a generalized framework for training tasks utilizing derivative loss components.

- Extend NTK to encompass this new paradigm from an analytical standpoint. Derive a kernel formulation from the convergent equation, akin to the NTK, which facilitates rigorous analysis.

- Introduce a geometric perspective on the interplay between any two loss components during the training process. This perspective offers insight into how models can converge consistently in both loss components prior to training.

## 2 PROPOSITIONS

In this section, we present a paradigm incorporating derivative labels within the context of the NTK. This proposition aims to unify and extend the applicability of these approaches. We omit depth of network for which isn't we care, and notice that all parameters $\theta$ in next article is defined as $\theta = (\theta_1^T, \theta_2^T, ..., \theta_n^T)$ in which $\theta_1$ denotes the i-th layer parameters.

### 2.1 THEORETICAL RESULTS

In this subsection, we leverage the invariance properties of the Neural Tangents Kernel (NTK) to analyze the convergence behavior in Stochastic Gradient Descent (SGD) when derivative labels are integrated into the loss function. To derive the convergent equation, we rely on the smooth assumption presented in Franceschi et al. (2022) and introduce the following proposition:

***Proposition* 1** We define two invariant about $\theta$ for the extra loss about two samples $x, x'$:

$$\hat{\Theta}'(x, x') \triangleq \nabla_\theta^T \partial_h f_t(x) \nabla_\theta f_t(x') \tag{6}$$

$$\hat{\Theta}''(x, x') \triangleq \nabla_\theta^T \partial_h f_t(x) \nabla_\theta \partial_h f_t(x') \tag{7}$$

which can be easily computed without iteration replaced by a simple differential operator (i.e. empirical). Notice that such kernel also converge to an expression of expectation as the width increasesArora et al. (2019a)Jacot et al. (2018). Induction is provided in Appendix A.1, and these invariants play a crucial role in our next proposition. Additionally, we offer two types of analyses for the convergence of equation (5): one follows the conventional approach of inducing a convergent equation in NTK, and the other involves a geometric analysis of the two loss components.

With perspective of induction, set $|Tr| = m$, we extend analysis of continuous-time gradient descent dynamics on (3) as following with omitting scale $\frac{1}{m}$:

$$\partial_t \theta = -\nabla_\theta L(Tr)$$
$$= \lambda_r \sum_{j=1}^m (\mathscr{L}(x_j) - f_t(x_j)) \nabla_\theta f_t(x_j) + \lambda_g \sum_{j=1}^m (\mathscr{L}'(x_j) - \partial_h f_t(x_j)) \nabla_\theta \partial_h f_t(x_j) \tag{8}$$

By chain rule, Network $f$ and its derivative structure evolves as:

$$\partial_t f_t(\boldsymbol{x}) = \partial_t \theta^T \nabla_\theta f_t(\boldsymbol{x})$$
$$\partial_t \partial_h f_t(\boldsymbol{x}) = \partial_t \theta^T \nabla_\theta \partial_h f_t(\boldsymbol{x}) \tag{9}$$

which can be induced and consist a system of differential equations about $f_t(\boldsymbol{x}), \partial_h f_t(\boldsymbol{x})$ as follows (detailed induction in Appendix A.2):

$$\begin{cases} \partial_t f_t(\boldsymbol{x}) + \lambda_g \hat{\Theta}'(\boldsymbol{x}, \boldsymbol{x}) \partial_h f_t(\boldsymbol{x}) = \lambda_r \hat{\Theta}(\boldsymbol{x}, \boldsymbol{x})(\mathscr{L}(\boldsymbol{x}) - f_t(\boldsymbol{x})) + \lambda_g \hat{\Theta}'(\boldsymbol{x}, \boldsymbol{x}) \mathscr{L}'(\boldsymbol{x}) \\ \partial_{th}^2 f_t(\boldsymbol{x}) + \lambda_g \hat{\Theta}''(\boldsymbol{x}, \boldsymbol{x}) \partial_h f_t(\boldsymbol{x}) = \lambda_r \hat{\Theta}'(\boldsymbol{x}, \boldsymbol{x})(\mathscr{L}(\boldsymbol{x}) - f_t(\boldsymbol{x})) + \lambda_g \hat{\Theta}''(\boldsymbol{x}, \boldsymbol{x}) \mathscr{L}'(\boldsymbol{x}) \end{cases} \quad (10)$$

in which $\boldsymbol{x}$ represents all the samples in the training set, and $h$ corresponds to the Gâteaux directions. As observed, when an additional loss is introduced into the loss function, the convergent equation concerning $t$ evolves into two equations. Specifically, at a certain time point $t_0$, $\partial_h f_t(\boldsymbol{x}, t_0)$ can be induced by $f_t(\boldsymbol{x}, t_0)$, as demonstrated in Rodini (2022). This induced value is related to both the network parameters $\theta$ and the input point $\boldsymbol{x}$ but not to the time variable $t$. While we assume that these quantities are independent of each other with respect to $t$; thus, it transforms into a typical set of ordinary differential equations with respect to $t$. We establish that such equations can be solved and evolve into a new kernel.

**Proposition 2** Solution of (11) is:

$$\begin{pmatrix} f_t(\boldsymbol{x}) \\ \partial_h f_t(\boldsymbol{x}) \end{pmatrix} = e^{-\hat{\Theta}^* t} \boldsymbol{C} + \begin{pmatrix} \mathscr{L} \\ \mathscr{L}' \end{pmatrix} \quad (11)$$

in which:

$$\hat{\Theta}^* = \begin{pmatrix} \lambda_r \hat{\Theta} & \lambda_g \hat{\Theta}' \\ \lambda_r \hat{\Theta}' & \lambda_g \hat{\Theta}'' \end{pmatrix}. \quad (12)$$

A detailed derivation is provided in Appendix A.2. As is well-known in NTK analysis, the convergence properties depend on the kernel $\hat{\Theta}$ and its eigenvalues. Given its positive nature, as $t$ increases, the function will tend to the objective function. In the case of training with first-order derivatives, the convergence properties will depend on $\hat{\Theta}^*$. This kernel is also semi-positive since it is a Gram matrix similar to NTK, constructed using the vector $(f^T(\boldsymbol{x}), \partial_h f^T(\boldsymbol{x}, t))^T$. Therefore, both $\partial_h f_t(\boldsymbol{x})$ and $f_t(\boldsymbol{x})$ converge to their respective labels.

Furthermore, in PINNWang et al. (2022), one can dynamically adjust the weights with respect to both loss components. In a generic PINN function, these weights change in each iteration, incurring additional computational overhead due to the kernel computations. However, as indicated in Proposition 1, we have the option to skip these adaptive weight iterations by initializing the weights, which theoretically takes a trade-off in both regular loss item and derivative loss item about convergence rate. In the following article, we will conduct experiments to verify such proposition.

**Proposition 3** Optimal Weight Setup:

$$\begin{cases} \lambda_r = \frac{tr(\hat{\Theta}*)}{tr(\hat{\Theta})} \\ \lambda_g = \frac{tr(\hat{\Theta}*)}{tr(\hat{\Theta}'')} \end{cases} \quad (13)$$

which we provide short description in Appendix A.3.

From a geometric perspective, we examine the parameter updates in equation (8). In each step, the direction of parameter updating depends on $\nabla_\theta f_t(x_j)$ and $\nabla_\theta \partial_{h_j} f_t(x_j)$ (or the opposite direction if the network's output is greater than its label) for any given sample. Consequently, the direction of parameter updating in (8) can be expressed as a linear combination of the vectors $\nabla_\theta f_t(x_j)$ and $\nabla_\theta \partial_h f_t(x_j)$ with corresponding coefficients $\mathscr{L}(x_j) - f_t(x_j)$ and $\mathscr{L}'(x_j) - \partial_h f_t(x_j)$. The angle between these two vectors signifies the degree of consistency between the two loss components, which remains invariant during training under the limitation of infinite-width network, as outlined below:

**Proposition 4** Angle $\alpha_j$ of two vectors $\nabla_\theta f_t(x_j)$ and $\nabla_\theta \partial_h f_t(x_j)$ are invariant on any given sample $x_j$ in the in the assumption that NTK is constant, and cosine of the angle is equal to:

$$cos\alpha_j = \frac{\hat{\Theta}'(x_j, x_j)}{\sqrt{\hat{\Theta}(x_j, x_j) * \hat{\Theta}''(x_j, x_j)}} \quad (14)$$

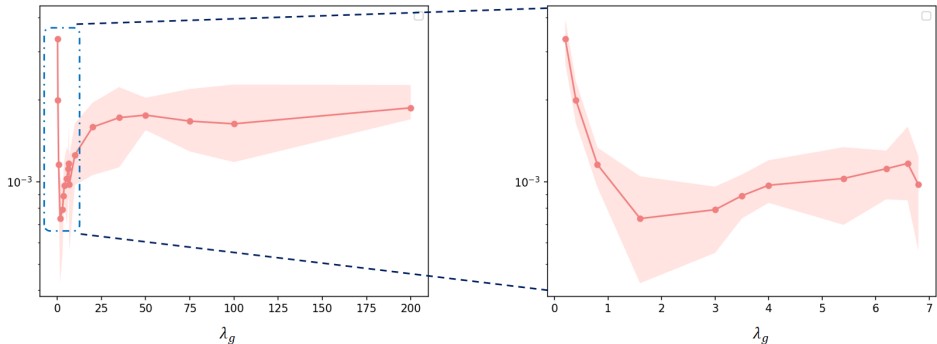

Figure 1: Experimental results for convergence of proposition 3, in which we always set $\lambda_r = 1$ and change $\lambda_g$, and we normalize $\lambda_r, \lambda_g$ to [0,1]. learning rate is 1e-3, epochs is 40000 with full batch training. The y-coordinate is $L^2$ loss.

it is obvious due to $\hat{\Theta}'(x_j, x_j)$ is actually inner product of $\nabla_\theta f_t(x_j)$ and $\nabla_\theta \partial_{h_j} f_t(x_j)$, $\hat{\Theta}(x_j, x_j), \hat{\Theta}''(x_j, x_j)$ are modules of both respectively. We notice readers that such invariant of angles is only be induced here for derivative loss but not other kind of multi-loss. Notice that in previous workFort et al. (2020), similar definition is used for kernel distance between different training steps while it's not same topic with ours.

## 2.2 EXPERIMENTAL CORROBORATION

We further verify our proposition by conducting experiments with fully connected ANNs. In experiments for proposition 1-3, we fit function $f_t(\boldsymbol{x}) = x_1 x_2$ in which $\boldsymbol{x} = (cos(\gamma), sin(\gamma))$, training set was generated by randomly selecting $\gamma$ in distribution $\mathcal{N}(0,1)$; in experiments for proposition 4, we conduct an extra experiment. In all our experiments, we set the width to 200 using a 4-layer FNN.

**Verification of Proposition 1**. We firstly verify our proposition 1 trained with normal least-squares cost and without derivative labels as Figure A1 in Appendix A.5 shows, which tends to be constant regardless of trained or not by increasing of width.

**Verification of Proposition 2**. $\hat{\Theta}', \hat{\Theta}''$ are same to NTK $\hat{\Theta}$ which tends to stable with width of network increasing. Moreover, by thearom in Du et al. (2018), the minimum eigenvalue $\lambda_{min}(\hat{\Theta})$ is a key to convergence rate of gradient flow. With empirical results that after adding derivative labels in tasks, accuracy in previous loss model will decrease and training convergence is affected, we utilize our proposition to explain this phenomenon and conduct experiment on that. Convergence in such kernel setting depends on $\lambda_0$, we shows that in same input and initialization, $\lambda_0$ of $\hat{\Theta}^*$ is less than $\hat{\Theta}$ ($\lambda_0(\hat{\Theta}^*) \approx 0.00662, \lambda_0(\hat{\Theta}) \approx 0.01157$), which leads to slow convergence. We implement 10 times with different training set and two loss function with and without derivative labels, namely, $f_1'(\boldsymbol{x}) = x_2$ and $f_2'(\boldsymbol{x}) = x_1$. Results are shown as Figure A2(a) in Appendix A.5.

As we can see, training with derivative labels is slower than without, which is caused by $\lambda_{min}(\hat{\Theta}^*) < \lambda_{min}(\hat{\Theta})$.

**Verification of Proposition 3**. This experiment is set to compare that where weights $\lambda_g, \lambda_r$ are best, we compute the theoretical setting is $\lambda_r \approx 1, \lambda_g \approx 3$, our setup is similar to Wang et al. (2022). While notice two points: in PINN, the second loss item represents physical information of desired function itself but in our paradigm it's perspective of interpolation that is theoretically same effect of the regular loss item; in PINN experiments, each experiments hold same learning rate and weights are set from [0,500] without normalization, which causes different learning rate in each step. Therefore, after selecting each weights, we normalize them two to [0,1] with summation 1. Results in Figure 1 shows that it's stronger evidence for our proposition 3, in which there are a obvious minimum around theoretical value 3.

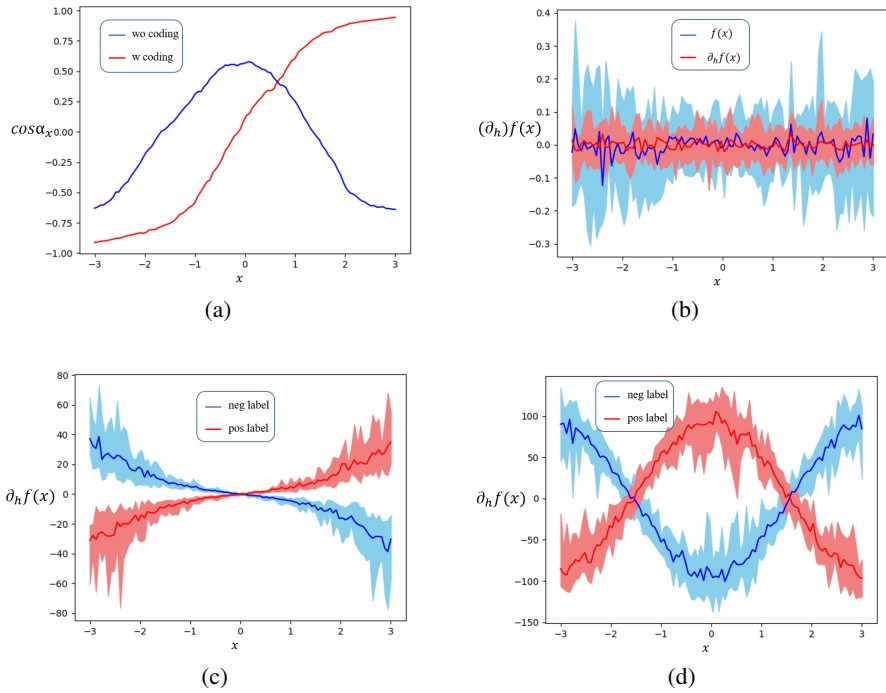

Figure 2: Experiments for proposition 4. We conduct two kinds of inputs: in 'wo coding' setup, inputs are uniformly select from $[-3, 3]$ as results shown in (c); in 'w coding' setup, inputs are replaced by fourier encoding as shown in (d). (a) angles of two kinds; (b) output $f(x)$ and its derivative $\partial_h f(x)$ after initialization (i.e. pre-activation); (c) derivatives of each training samples with two kinds of labels without encoding training; (d) derivatives of each training samples with two kinds of labels with encoding training

**Verification of Proposition 4**. We firstly verify our proposition 4 by recording angle of each samples in training process, we find that its fluctuation is actually very tiny with 1e-3 variants. Novak et al. (2018) and Lee et al. (2017) has shown that training a certain point makes derivative in the point increases in early period. We observed that derivative variants during the training process, even in regular training, are associated with the angle of points. To verify this hypothesis, we conducted experiments aimed at eliminating unrelated factors. We trained a single point through several steps with only regular loss item (that is, without derivative item in training), always setting the label from $10, -10$ until the output of such a point larger ( smaller) than 10(-10).

At initialization, it's well-known that the outputs of samples are close to zero. We also found that their derivatives are similarly close to zero, as depicted in Figure 2(b). We tested two types of inputs: those with position encoding and those without. For inputs with position encoding, we replaced our inputs with $\boldsymbol{x_e} = cos(x), sin(x)$, where scalar inputs $x$ were uniformly selected from [-3,3]. After several training iterations, we observed that their derivatives were predicted by our angle, as illustrated in Figure 2(a,c,d). In Figure 2(a), we display the angles for the two types of input, while (c) and (d) represent the two types within two different sets of labels.

## 2.3 GENERALITY

We have only discussed single extra one-order loss item above, in this section, we provide a general formula. As extending (5):

$$L(\boldsymbol{x}) \triangleq \frac{1}{2} \sum_{i=0}^{k} \sum_{j=1}^{l_i} \lambda_{ij} ||\partial_{b_{ij}}^{(i)} f_t(\boldsymbol{x}) - \mathscr{L}_{b_{ij}}^{(i)}(\boldsymbol{x})||_2^2, \quad b_{ij} = \{h_1^{(i)}, h_2^{(i)}, ..., h_{l_i}^{(i)}\} \tag{15}$$

in which we define $\partial^{(0)} f_t(x) = f_t(x)$, we extend all previous propositions with inductions shown in Appendix A.4. In the next section, we demonstrate that our interpolation paradigm contains Jacobian Regularization, and our geometric perspective can be used also in PINN.

## 3 TYPICAL EXAMPLES

In this section, we discuss several tasks that is trained with derivative labels. In the following, the first two tasks: PINN and image classification obtain experimental results in previous work, and we discuss batch setting with our geometric perspective in neural network finally.

### 3.1 PINN

We consider PDEs in PINN, in which network is used to fit a certain function, input norm space $X$ is $R^n$, the fitted function should be smooth (i.e. $f \in C^\infty[a,b]$, $a, b$ is considered range for the problem). As aforementioned, convergence properties are revealed with perspective of NTK in Wang et al. (2022). Such formula cannot be described within our labeled method. However, we can still consider geometrical property of general first order homogeneous linear partial differential equation $f_t(x_1, x_2, ..., x_n)$:

$$\sum_{i=1}^{n} g_i(x_1, x_2, ..., x_n) \partial_{x_i} f = 0 \tag{16}$$

which contains various significant equations like EE(eular equation)Mao et al. (2020), such kind of PDEs plays role of extra loss item in PINN with square, which is the derivative loss item besides the loss of fitting error. direction of parameters updating is:

$$\partial_t \theta = \sum_{i=1}^{n} g_i(x_1, x_2, ..., x_n) \partial_{x_i} f * \sum_{i=1}^{n} g_i(x_1, x_2, ..., x_n) \nabla_\theta \partial_{x_i} f \tag{17}$$

The angle of two loss item of regular and derivative is similarly:

$$cos\alpha = \frac{\nabla_\theta^T f \sum_{i=1}^{n} g_i(x_1, ..., x_n) \nabla_\theta \partial_{x_i} f}{||\nabla_\theta f|| * ||\sum_{i=1}^{n} g_i(x_1, ..., x_n) \nabla_\theta \partial_{x_i} f||} \tag{18}$$

in which we use $\hat{\Theta}'_{x_i}, \hat{\Theta}''_{x_i x_j}$ to denote $\nabla_\theta^T f \nabla_\theta \partial_{x_i} f$ and $\nabla_\theta^T \partial_{x_i} f \nabla_\theta \partial_{x_j} f$ respectively, that angle is constant for certain sample:

$$cos\alpha = \frac{\sum_{i=1}^{n} g_i(x_1, ..., x_n) \hat{\Theta}'_{x_i}}{\hat{\Theta} * \sum_{i=1}^{n} \sum_{j=1}^{n} g_i(x_1, ..., x_n) g_j(x_1, ..., x_n) \hat{\Theta}_{x_i x_j}} \tag{19}$$

### 3.2 IMAGE CLASSIFICATION

In this subsection, we consider the specific task in image classification of its robustness for defending adversarial attack, which is implemented in with Jacobian Regularization, our paradigm is equivalent to such regularization for its ideal derivative label is actually zero. In image classification, input norm space $X$ is specifically $R^{m \times n}$, we consider a usual Frobenius norm $|| \cdot ||_F$ in such space, the Gâteaux Derivative is some direction $h \in R^{m \times n}$ denoting a image noise direction. For example, for a pixel-wise and pairwise independent noise attacked, direction $h$ should be a quantity matrix. An ideal situation is that whatever kind of noise is added on image, network is robust to correctly predict, which implies that whatever $h$ is, the one-order Gâteaux derivative is zero. Formally, for any $h \in R^{m \times n}, x \in Tr$, zero value of (1) is as one-order label of such task (i.e. $\mathscr{L}'(x) = 0$). That means Gâteaux Derivative of each bases in norm space $R^{m \times n}$ is zero, which is the usual partial derivative about each dimension, that is equivalent to Jacobian Regularization, consider flatten vector of image $x = \{x_i\}_{m \times n}$, that (4) is:

$$L(x) = \frac{1}{2}[f_t(x) - \mathscr{L}(x)]^2 + w \sum_{i=1}^{m \times n} \frac{1}{2}[\partial_{x_i} f_t(x) - 0]^2$$
$$= \frac{1}{2}[f_t(x) - \mathscr{L}(x)]^2 + \frac{1}{2}w||\partial_x f_t(x)||_F^2 \tag{20}$$

Thanks to Arora et al. (2019a), one can compute NTK for CNN with or without average pooling, which represents that such Regularization can be involved in our propositions.

### 3.3 DISCUSSION IN BATCH

In this subsection, we discuss batch setup in neural network by extending our proposition 4 to regular item. As known that a regular task with batch is as:

$$L(\boldsymbol{x}) \triangleq \frac{1}{2}||f_t(\boldsymbol{x}) - \mathscr{L}(\boldsymbol{x})||_2^2 \tag{21}$$

and similar to above, its parameters updating flow is:

$$\partial_t \theta = \sum_{i=1}^{m} (\mathscr{L}(x_i) - f_t(x_i)) \nabla_\theta f(x_i) \tag{22}$$

which is linear combination of vectors $\{\nabla_\theta f(x_i)\}_{i=1,2,\dots,m}$. Then we can extend proposition 4 as follows:

**Proposition$_*$ 4** Angle $\alpha_{ij}$ of two samples $\{x_i, x_j\}$ within parameters updating is as:

$$cos\alpha_{ij} = \frac{\hat{\Theta}(x_i, x_j)}{\sqrt{\hat{\Theta}(x_i, x_i) * \hat{\Theta}(x_j, x_j)}} \tag{23}$$

Empirical evidence suggests that randomly and uniformly selecting batches is more effective, particularly in coordinate input scenariosMildenhall et al. (2021), where samples are not discrete from each other. Uniform selection helps the model maintain a better representation of the data distribution. We propose that for a given network architecture, all samples become correlated with each other after inference, and the degree of correlation is indicated by the angle mentioned earlier. While in the training process, direction of parameters may inverse, which leads to turbulence of training loss. We conjecture that while in orthogonal, it's most stable for training loss for reason that direction inverse will not change their angle.

We conducted experiments based on this concept, as illustrated in Appendix A.5. In Figure A2(b), we trained a function $f(\boldsymbol{x}) = \sin(x_1)\cos(x_2)$ with a batch size of 2 with four batch settings: 'normal', 'adaptive', 'worst', 'orthogo'. In the 'normal' setup, we randomly and uniformly selected the second sample; in the 'adaptive' setup, we chose the sample that is farthest away (with the lowest cosine value) from the first sample; in the 'worst' setup, we chose the sample that is nearest away (with the lowest cosine value) from the first sample; in the 'orthogo' setup, we chose the sample that is orthogonal from the first sample . In each setup, we iterated through all training samples. Results shows that 'orthogo' holds better convergence than others. About the results, we propose two conjectures: **I**. In the batch setting, samples with small angle hurt training generality, which makes such training results of batch weak generalization of full training set; **II**. In the training process, parameter updating directions will change, whose results may cause turbulence of training. Why is 'orthogonal' training superior? With respect to **I**, it helps maintain generality by considering the perspective of updating directions. Regarding Conjecture **II**, it preserves the angle during the training process, even if the direction is inverted.

## 4 RELATED WORK

### 4.1 NEURAL TANGENT KERNEL

The Neural Tangent Kernel (NTK), a widely adopted gradient kernel, was initially introduced by Jacot Jacot et al. (2018). This kernel takes the form of a Gram matrix constructed from gradients and offers an equivalence to training a fully-connected neural network under specific parameterization, resembling a kernel method, whose potential of infinite-based is illustrated in the previous Lee et al. (2020). Over time, its applicability has been extended to various neural network architectures, including convolutional networks Samarin et al. (2020) Arora et al. (2019a), residual networks Yang & Schoenholz (2017), networks incorporating attention mechanisms Hron et al. (2020), analysis for GAN architecture Franceschi et al. (2022), even for graph neural networksDu et al. (2019).

The precise computation of this kernel is outlined in Arora et al. (2019a), as infinite network is ideal, the finite-width kernel was induced and able to compute similarlyNovak et al. (2022) while implementation details are provided in Engel et al. (2022). Such kernels play a pivotal role in convergence analysis, particularly in convergence rate evaluation, hinging on the minimum eigenvalue of the NTK. The applications of NTK have broadened significantly since its inception. With its constant kernel, one can readily utilize its convergence function to estimate the performance of a specific architecture. Notably, the NTK, functioning as a kernel, seamlessly integrates with most kernel methods Arora et al. (2019b).

Due to its capacity to encapsulate essential attributes of network architectures and datasets, the NTK has naturally found its way into Neural Architecture Search (NAS) Chen et al. (2021). This integration contributes to reducing time overhead within NAS Xu et al. (2021) Mok et al. (2022). Furthermore, the NTK holds substantial utility as a theoretical tool for analyzing input encoding in Coordinate-MLP Tancik et al. (2020) and convergence property of PINN Wang et al. (2022), explaining effective of adversarial training strategy Tsilivis & Kempe (2022) and active learning strategy Wang et al. (2021b) Mohamadi et al. (2022) Its applications extend to proof of zero training loss in polynomial time for over-parameterized residual networks Du et al. (2019), further underlining its significance.

## 4.2 TASKS WITH DERIVATIVE LOSS

In the field of interpolation, both samples and their derivatives find utility in model interpolation. Recently, the interplay between model robustness and the scale of parameters has garnered attention Bubeck et al. (2021) Bubeck & Sellke (2021). These works leverage the Lipschitz condition to articulate robustness. Notably, the bounded derivative of models serves as a less stringent robustness criterion, a notion also explored in Hoffman et al. (2019). However, in the context of neural networks, analyzing derivatives is complex due to their intricate expressions Rodini (2022).

Moreover, various tasks incorporate the gradient or derivative of predictions with respect to inputs as an additional loss component. For instance, a norm of a Jacobian Matrix Novak et al. (2018) is employed as regularization to enhance network robustness. In the realm of mesh reconstruction, SDF (Signed Distance Function) Wang et al. (2021a) incorporates an inductive differential function of inputs and outputs. More recently, Physical Informed Neural Networks (PINN) have emerged as a novel paradigm to tackle problems involving Partial Differential Equations (PDEs) Raissi et al. (2019) Cuomo et al. (2022). In PINN, the loss function encompasses an extra term comprising several partial derivatives. This paradigm facilitates the application of scientific methodologies to traditional numerical fields, particularly in equation solving, parameter inversion, model discovery, control, and optimization.

## 5 CONCLUSION

As the Neural Tangent Kernel has emerged as a powerful theoretical tool, it has been instrumental in explaining various phenomena, albeit primarily within tasks trained using the least-square loss. In this article, we consider a specific multi-loss, derivative loss in neural network, extending the analytical insights derived from the Neural Tangent Kernel to expound upon a broader class of loss functions, encompassing convergent ordinary differential equations, such extending provides kernel for discussed multi-loss and explain why tasks with derivative loss holds slow convergence, and we extended adaptive weights strategy in PINN to our discussion, which provides a strong selection in all application who use derivative loss item. Furthermore, we introduce a geometric perspective between any two loss components in neural network (in this article, two samples in batch, one samples with regular and derivative loss are discussed). This perspective offers a unique understanding of the intricacies involved and sheds light on potential relationship between loss component. To illustrate the versatility, we conduct experiments to illustrate our assumption and propositions. Our propositions reveal loss relationship of various kinds of two given components.

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

# A  APPENDIX

## A.1  PROOF OF PROPOSITION 1

As is known that continuous partial derivatives can be computatively ordered, we ignore discontinuous cases such as ReLu activation function for which can be replaced with continuous function with smooth assumptions by Franceschi et al. (2022). Therefore:

$$\hat{\Theta}'(x, x') = \nabla_\theta^T \partial_h f_t(x) \nabla_\theta f_t(x') = \partial_{h_x} \hat{\Theta}(x, x')$$

$$\hat{\Theta}''(x, x') = \nabla_\theta^T \partial_h f_t(x) \nabla_\theta \partial_h f_t(x') = \partial_{h_{x'}} \hat{\Theta}'(x, x')$$

in which $h_x = h_{x'} = h$, $h_x, h_{x'}$ denotes partial direction for $x, x'$ respectively.

## A.2  INDUCTION OF PROPOSITION 2

We firstly give induction of convergent function, recall that in content parameters flow as:

$$\partial_t \theta = \lambda_r \sum_{j=1}^m (\mathscr{L}(x_j) - f_t(x_j)) \nabla_\theta f_t(x_j) + \lambda_g \sum_{j=1}^m (\mathscr{L}'(x_j) - \partial_h f_t(x_j)) \nabla_\theta \partial_h f_t(x_j) \quad \text{(A-1)}$$

Network $f$ on a certain data point $x_0 \in X$ evolves as:

$$\begin{aligned}
\partial_t f_t(x_0) &= \partial_t \theta^T \nabla_\theta f_t(x_0) \\
&= \lambda_r \sum_{j=1}^m (\mathscr{L}(x_j) - f_t(x_j)) \nabla_\theta^T f_t(x_j) \nabla_\theta f_t(x_0) \ + \\
&\quad \lambda_g \sum_{j=1}^m (\mathscr{L}'(x_j) - \partial_{h_j} f_t(x_j)) \nabla_\theta^T \partial_h f_t(x_j) \nabla_\theta f_t(x_0) \\
&= \lambda_r \hat{\Theta}(x_0, \boldsymbol{x})(\mathscr{L}(\boldsymbol{x}) - f_t(\boldsymbol{x})) + \lambda_g \hat{\Theta}'(x_0, \boldsymbol{x})(\mathscr{L}'(\boldsymbol{x}) - \partial_h f_t(\boldsymbol{x}))
\end{aligned} \quad \text{(A-2)}$$

which denotes:

$$\partial_t f_t(x_0) + \lambda_g \hat{\Theta}'(x_0, \boldsymbol{x}) \partial_h f_t(\boldsymbol{x}) = \lambda_r \hat{\Theta}(x_0, \boldsymbol{x})(\mathscr{L}(\boldsymbol{x}) - f_t(\boldsymbol{x})) + \lambda_g \hat{\Theta}'(x_0, \boldsymbol{x})\mathscr{L}'(\boldsymbol{x}) \quad \text{(A-3)}$$

Consider on all the training set:

$$\partial_t f_t(\boldsymbol{x}) + \lambda_g \hat{\Theta}'(\boldsymbol{x}, \boldsymbol{x}) \partial_h f_t(\boldsymbol{x}) = \lambda_r \hat{\Theta}(\boldsymbol{x}, \boldsymbol{x})(\mathscr{L}(\boldsymbol{x}) - f_t(\boldsymbol{x})) + \lambda_g \hat{\Theta}'(\boldsymbol{x}, \boldsymbol{x})\mathscr{L}'(\boldsymbol{x}) \quad \text{(A-4)}$$

Such convergent equation is related to two different functions $f_t(\boldsymbol{x}), \partial_h f_t(\boldsymbol{x})$ about time $t$, for that we need extra equation about one-order derivative:

$$\begin{aligned}
\partial_t \partial_h f_t(x_0) &= \partial_t \theta^T \nabla_\theta \partial_h f_t(x_0) \\
&= \lambda_r \hat{\Theta}'(x_0, \boldsymbol{x})(\mathscr{L}(\boldsymbol{x}) - f_t(\boldsymbol{x})) + \lambda_g \hat{\Theta}''(x_0, \boldsymbol{x})(\mathscr{L}'(\boldsymbol{x}) - \partial_h f_t(\boldsymbol{x}))
\end{aligned} \quad \text{(A-5)}$$

similarly:

$$\partial_t \partial_h f_t(\boldsymbol{x}) + \lambda_g \hat{\Theta}''(\boldsymbol{x}, \boldsymbol{x}) \partial_h f_t(\boldsymbol{x}) = \lambda_r \hat{\Theta}'(\boldsymbol{x}, \boldsymbol{x})(\mathscr{L}(\boldsymbol{x}) - f_t(\boldsymbol{x})) + \lambda_g \hat{\Theta}''(\boldsymbol{x}, \boldsymbol{x})\mathscr{L}'(\boldsymbol{x}) \quad \text{(A-6)}$$

which consist a system of differential equations about $f_t(\boldsymbol{x}, t), \partial_h f_t(\boldsymbol{x}, t)$ as follows:

$$\begin{cases}
\partial_t f_t(\boldsymbol{x}) + \lambda_g \hat{\Theta}'(\boldsymbol{x}, \boldsymbol{x}) \partial_h f_t(\boldsymbol{x}) = \lambda_r \hat{\Theta}(\boldsymbol{x}, \boldsymbol{x})(\mathscr{L}(\boldsymbol{x}) - f_t(\boldsymbol{x})) + \lambda_g \hat{\Theta}'(\boldsymbol{x}, \boldsymbol{x})\mathscr{L}'(\boldsymbol{x}) \\
\partial_t \partial_h f_t(\boldsymbol{x}) + \lambda_g \hat{\Theta}''(\boldsymbol{x}, \boldsymbol{x}) \partial_h f_t(\boldsymbol{x}) = \lambda_r \hat{\Theta}'(\boldsymbol{x}, \boldsymbol{x})(\mathscr{L}(\boldsymbol{x}) - f_t(\boldsymbol{x})) + \lambda_g \hat{\Theta}''(\boldsymbol{x}, \boldsymbol{x})\mathscr{L}'(\boldsymbol{x})
\end{cases} \quad \text{(A-7)}$$

We skip the script $\boldsymbol{x}$ and denote $f_t(\boldsymbol{x})$ and $\partial_h f_t(\boldsymbol{x})$ as $Y(t)$ and $Z(t)$ respectively, $Y'(t)$ denotes $\partial_t f_t(\boldsymbol{x}, t)$ and $Z'(t)$ is similar, rewriting (A-7) as:

$$Y'(t) = -\hat{\Theta} Y(t) - \hat{\Theta}' Z(t) + (\hat{\Theta}\mathscr{L} + \hat{\Theta}'\mathscr{L}') \tag{A-8}$$

$$Z'(t) = -\hat{\Theta}' Y(t) - \hat{\Theta}'' Z(t) + (\hat{\Theta}'\mathscr{L} + \hat{\Theta}''\mathscr{L}') \tag{A-9}$$

which is a system of ordinary differential equations, with basic solution matrix:

$$-\hat{\Theta}^* = -\begin{pmatrix} \hat{\Theta} & \hat{\Theta}' \\ \hat{\Theta}' & \hat{\Theta}'' \end{pmatrix}.$$

Thus we have it's general solution, then we induce its particular solution. With eliminating variable $Y(t)$ and $Z(t)$ respectively, we can induce them as following: by (A-8), we have:

$$Z(t) = \hat{\Theta}'^{-1}[-Y'(t) - \hat{\Theta} Y(t) + (\hat{\Theta}\mathscr{L} + \hat{\Theta}'\mathscr{L}')] \tag{A-10}$$

then differentiate the two sides in equation (A-8):

$$Y''(t) = -\hat{\Theta} Y'(t) - \hat{\Theta}' Z'(t) \tag{A-11}$$

take (A-8) and (A-9) into (A-11) we have:

$$\begin{aligned} Y''(t) &= -\hat{\Theta} Y'(t) - \hat{\Theta}'\{-\hat{\Theta}' Y(t) - \hat{\Theta}''\hat{\Theta}'^{-1}[-Y'(t) \\ &\quad - \hat{\Theta} Y(t) + (\hat{\Theta}\mathscr{L} + \hat{\Theta}'\mathscr{L}')] + (\hat{\Theta}'\mathscr{L} + \hat{\Theta}''\mathscr{L}')\} \\ &= -(\hat{\Theta} + \hat{\Theta}'\hat{\Theta}''\hat{\Theta}'^{-1})Y'(t) + (\hat{\Theta}'^2 - \hat{\Theta}'\hat{\Theta}''\hat{\Theta}'^{-1}\hat{\Theta})Y(t) \\ &\quad + \hat{\Theta}'\hat{\Theta}''\hat{\Theta}'^{-1}(\hat{\Theta}\mathscr{L} + \hat{\Theta}'\mathscr{L}') - \hat{\Theta}'(\hat{\Theta}'\mathscr{L} + \hat{\Theta}''\mathscr{L}') \end{aligned}$$

As all $\hat{\Theta}, \hat{\Theta}', \hat{\Theta}''$ are invertible and positive symmetry, matrix multiplication among them is exchangeable, therefore:

$$\begin{aligned} Y''(t) &= -(\hat{\Theta} + \hat{\Theta}'')Y'(t) + (\hat{\Theta}'^2 - \hat{\Theta}\hat{\Theta}'')Y(t) \\ &\quad + \hat{\Theta}''(\hat{\Theta}\mathscr{L} + \hat{\Theta}'\mathscr{L}') - \hat{\Theta}'(\hat{\Theta}'\mathscr{L} + \hat{\Theta}''\mathscr{L}') \\ &= -(\hat{\Theta} + \hat{\Theta}'')Y'(t) + (\hat{\Theta}'^2 - \hat{\Theta}\hat{\Theta}'')Y(t) + (\hat{\Theta}\hat{\Theta}'' - \hat{\Theta}'^2)\mathscr{L} \end{aligned}$$

similar to function $Z(t)$, we induce two independent equation about $Y(t), Z(t)$ respectively:

$$Y''(t) + (\hat{\Theta} + \hat{\Theta}'')Y'(t) + (\hat{\Theta}\hat{\Theta}'' - \hat{\Theta}'^2)Y(t) = (\hat{\Theta}\hat{\Theta}'' - \hat{\Theta}'^2)\mathscr{L}$$

$$Z''(t) + (\hat{\Theta} + \hat{\Theta}'')Z'(t) + (\hat{\Theta}\hat{\Theta}'' - \hat{\Theta}'^2)Z(t) = (\hat{\Theta}\hat{\Theta}'' - \hat{\Theta}'^2)\mathscr{L}'$$

It's obvious that particular solution is constant:

$$Y^*(t) = \mathscr{L}, \quad Z^*(t) = \mathscr{L}' \tag{A-12}$$

## A.3 INDUCTION OF PROPOSITION 3

As proposition 2 shows a general solution system of ordinary differential equations, whose scalar formulation is related about eigenvalues and eigenvectors of matrix $\hat{\Theta}$ (here we use 'matrix' instead 'kernel' to denote $\hat{\Theta}$ appropriately). As we know as real symmetric matrix, there are only real eigenvalues and eigenvectors. We define the average convergence representation as sum of eigenvalues:

$$\lambda^* = tr(\hat{\Theta}^*) \tag{A-13}$$

so that we can see in proposition 3 setting, the average convergence representation doesn't change but the regular average convergence representation and derivative correspondence is equal:

$$\begin{aligned} tr(\lambda_r \hat{\Theta}) &= tr(\lambda_g \hat{\Theta}'') \\ tr(\lambda_r \hat{\Theta}) + tr(\lambda_g \hat{\Theta}'') &= tr(\hat{\Theta}^*) \end{aligned} \tag{A-14}$$

## A.4 GENERALITY

For concisely showing induction, we skip order-time subscript, and define that $\{h_i\}_{i \in \{1,2,..,k\}}$ is a set of derivative subscript set with arbitrary length. Moreover, we demand several order smoothness for NTK, as Assumption 1.

**Proposition\* 1** We define:

$$\hat{\Theta}_{h_i h_j}(x, x') \triangleq \nabla_\theta^T \partial_{h_i}^{|h_i|} f_t(x) \nabla_\theta \partial_{h_j}^{|h_j|} f_t(x') \tag{A-15}$$

that is constant in infinite neural network, this proof is same to Proposition 1.

**Proposition\* 2** Solution of (11) is:

$$\begin{pmatrix} \partial_{h_1}^{|h_1|} f_t(\boldsymbol{x}) \\ \partial_{h_2}^{|h_2|} f_t(\boldsymbol{x}) \\ ... \\ \partial_{h_k}^{|h_k|} f_t(\boldsymbol{x}) \end{pmatrix} = e^{-\hat{\Theta}^* t} \boldsymbol{C} + \begin{pmatrix} \mathscr{L}_{h_1} \\ \mathscr{L}_{h_2} \\ ... \\ \mathscr{L}_{h_k} \end{pmatrix} \tag{A-16}$$

in which $\hat{\Theta}^*$ is a $km \times km$ matrix shown in $k \times k$ chunked matrix, its (i,j)th chunked entry is:

$$\hat{\Theta}_{ij}^* = (\lambda_{ij} \hat{\Theta}_{h_i h_j})_{k \times k} \tag{A-17}$$

**Proof**: We firstly give induction of each convergent function, parameters flow as:

$$\partial_t \theta = \sum_{i=1}^k \lambda_i \sum_{j=1}^m (\mathscr{L}_{h_i}(x_j) - \partial_{h_i}^{|h_i|} f_t(x_j)) \nabla_\theta \partial_{h_i}^{|h_i|} f_t(x_j) \tag{A-18}$$

A certain derivative item $h_a$ on a certain data point $x_0 \in X$ evolves as:

$$\begin{aligned} \partial_t \partial_{h_a}^{|h_a|} f_t(x_0) &= \partial_t \theta^T \nabla_\theta \partial_{h_a}^{|h_a|} f_t(x_0) \\ &= \sum_{i=1}^k \lambda_i \sum_{j=1}^m (\mathscr{L}_{h_i}(x_j) - \partial_{h_i}^{|h_i|} f_t(x_j)) \nabla_\theta^T \partial_{h_i}^{|h_i|} f_t(x_j) \nabla_\theta \partial_{h_a}^{|h_a|} f_t(x_0) \\ &= \sum_{i=1}^k \lambda_i \hat{\Theta}_{h_i h_a}(x_0, \boldsymbol{x})(\mathscr{L}_{h_i}(\boldsymbol{x}) - \partial_{h_i}^{|h_i|} f_t(\boldsymbol{x})) \end{aligned} \tag{A-19}$$

On the full training set, this construct a system of ordinary differential equations for all $h_a$. Its basic solution matrix and particular solution is obvious.

**Proposition\* 3** Optimal Weight Setup:

$$\lambda_i = \frac{tr(\hat{\Theta}^*)}{tr(\hat{\Theta}_{h_i h_i})} \tag{A-20}$$

**Proposition\* 4** Angle $\alpha_l^{(ij)}$ of two loss items for a certain point $x_l$ is similarly as:

$$cos\alpha_l^{(ij)} = \frac{\hat{\Theta}_{h_i h_j}(x_l, x_l)}{\sqrt{\hat{\Theta}_{h_i h_i}(x_l, x_l) * \hat{\Theta}_{h_j h_j}(x_l, x_l)}} \tag{A-21}$$

these generalities allow us to explain most of neural tasks with derivative loss items including image processing, PINN and so on.

## A.5 EXTRA FIGURE

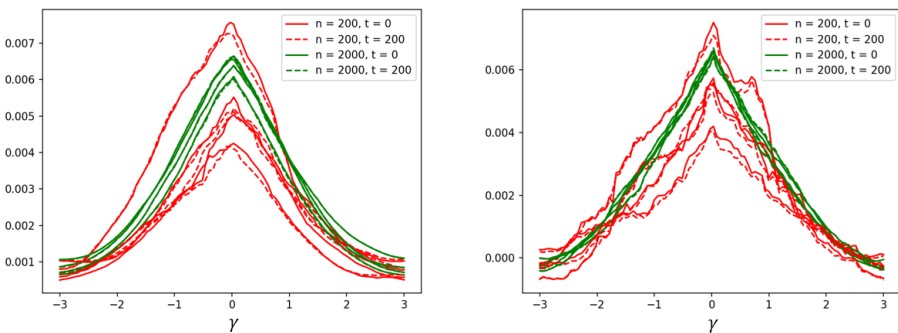

Figure A1: Experimental results for convergence of proposition 1 to a fixed limit for two widths n and two times t. (a), (b) denotes $\hat{\Theta}', \hat{\Theta}''$ respectively in which we performed three independent initialization of network weights and trained respectively

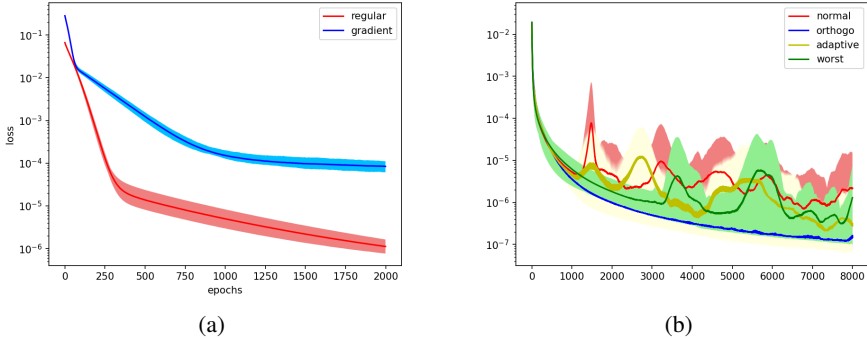

Figure A2: Experimental results for verification, (a) in 'gradient', we set weight $\lambda_r = \lambda_g = 1$ with fitting $f_t(\boldsymbol{x}) = x_1 x_2$ in which $\boldsymbol{x} = (cos(\gamma), sin(\gamma))$; while in 'regular', $\lambda_r = 1$ and $\lambda_g = 0$. Results shows that 'regular' converge slower than 'gradient'. (b) we conduct 10 times for two kind of batch setting, plot their average lines with their upper and lower confident bounds;

