# OpenReview forum: "How Neural Networks With Derivative Labels Work: A Neural Tangent Kernel Perspective"
_ICLR.cc/2024/Conference — Submitted to ICLR 2024_

### Official Review · Reviewer_PpX8 · 2023-10-27

**Soundness:** 2 fair
**Presentation:** 2 fair
**Contribution:** 2 fair
**Rating:** 3
**Confidence:** 3

**Summary:**

This paper builds on the NTK framework, and proposes Gateaux derivatives as a method to include additional loss terms like Jacobian regularization and PINN. This paper does not have empirical validations of their theory.

**Strengths:**

This paper aims to extend the NTK framework to include auxiliary loss terms like Jacobian regularization and PINN.

**Weaknesses:**

The main weakness of this paper is that the authors did not discuss any implications or potentials of their proposed change to the NTK framework. Will this paper allow us to better understand the NTK framework? Will the analysis in this paper allow us to better train neural networks?

**Questions:**

See Weakness

---

### Official Review · Reviewer_RXR5 · 2023-11-01

**Soundness:** 2 fair
**Presentation:** 1 poor
**Contribution:** 2 fair
**Rating:** 3
**Confidence:** 5

**Summary:**

The authors attempt to extend the NTK approach to optimization settings where the loss includes model gradient terms in addition to the standard MSE terms. They show how this extension leads to analogous results as an attempt to provide a theory for optimization in the extended setting and provide numerical experiments to test and support their claims.

**Strengths:**

The authors are tackling a novel and substantial direction of modern ML research, while still extending from and using tools that are well understood within the community. While the NTK is obviously a significant advance in building theories for ML optimization, it has also suffered from being applicable in a too narrow setting and the authors' attempt to find an extension in cases of the losses that are of generic interest is commendable.

**Weaknesses:**

A. The work suffers a lot in terms of readability due to poor grammatical and mathematical presentation ("Neural $Tangents$ Kernel", Proposition 2 saying "Solution of (11) is (11)", etc). $\Theta '$ is said to be symmetric, but there seems to be no obvious reason how that can be a simple statement that doesn't need proof ($\Theta ''$ being symmetric is more acceptable). Similarly, the proof of Proposition * 1 is said to be the same as the proof given in Proposition 1, but that proof effectively just restates the proposition from the main text.

These and many other examples are strong flaws that will impact readability substantially. Regarding grammar alone, in an age where grammatical tools are easily and freely available, I would like to strongly suggest that the authors take the time to run their text through these kinds of tools (and apologize in advance for any grammatical errors I might have made in my own review).

B. I am not sure if the handwaved switches between stochastic gradient descent and deterministic gradient descent (over which NTK analysis is usually rooted in) are valid (further, for the purposes of developing the theory they are also perhaps unneeded). Where is the proof that the newer NTK like objects $\Theta', \Theta''$... retain the static kernel property as well (or a proof sketch on why Jacot's approach should extend seamlessly to these new objects to obtain such a result)? Similarly, the Gateux derivative is certainly not an element of the reals, as is written just after Eq. 1 (the reals contain the range of that operator). PINNs are mentioned multiple times as providing validation for the approach, but the numerical evidence of that claim is scant. The mathematical structure being invoked from functional analysis is notoriously sensitive to even infinitesmial changes, but is treated carelessly in the text.

I find the work's ambition to be impressive, but the handwaviness of the mathematical treatment leaves me with no confidence that the stated results stand rigorously (even though, on some level they appeal to intuition). The numerical experiments are not impressive enough to consider them as sufficiently compensating for that fact either. As such, I will have to keep my first score at a reject (but I am happy to engage with the authors on why my observations may have failed me and open to the possibility of a substantial change in score).

**Questions:**

NA

---

### Official Review · Reviewer_LVg4 · 2023-11-04

**Soundness:** 2 fair
**Presentation:** 2 fair
**Contribution:** 2 fair
**Rating:** 3
**Confidence:** 3

**Summary:**

Neural Tangent Kernel (NTK) analyzes the convergence and generalization of deep neural networks. NTK for networks trained with additional derivative loss items like Jacobian Regularization is challenging. This paper proposes a paradigm that uses Gateaux derivative labels to extend NTK. The authors further give the derivation from two perspectives and then verify them through experiments.

**Strengths:**

- Although the presentation of this paper needs further improvement, it is well organized with empirical support.

- This paper provides a unified explanation of the NTK with additive derivative loss items. Such an extension can expand the utility of NTK to more diverse tasks.

- This paper provides both inductive and geometrical perspectives, which gives a comprehensive understanding of the hard convergence phenomenon.

**Weaknesses:**

- The literature summary is not good, and some statements can be misleading. For example, the author claims that "However, its convergent equation is currently limited to the least-square loss, whereas multi-loss training is a widely employed approach in practical applications." However, NTK-type analysis is not limited to least-square loss. Other types of loss functions, such as cross-entropy loss, have also been analyzed in numerous literature.

- This paper proposes a new method but doesn't apply it to real data sets. Only simple synthetic data distributions are considered.

- This paper hasn't considered the algorithm. It is unclear how to implement the idea in this paper to real data sets. This will affect the impact of this paper. NTK-type analysis requires the weight to be close to the initialization. It is also unclear whether the additional derivative loss will drive the weight far from the initialization.

- This paper only considers the infinite neural network. It is unclear what the additional derivative loss items look like in the finite width condition."

**Questions:**

- Is the method applicable to non-smooth activation functions like ReLU? While the original NTK can deal with ReLU, I'm not sure how it does for additional derivative loss items.

- Is there a general algorithm presented in this paper about how to train such a model? I looked into the code provided by the author, which is very specific to the synthetic distribution."

---

### Meta-Review · Area_Chair_wJo1 · 2023-12-10

**Metareview:**

This paper generalizes the Neural Tangent Kernel (NTK) framework to include an additional derivative loss item. It was clear to all the reviewers that this paper was not ready for publication given its many issues. No author rebuttal was provided.

**Justification For Why Not Higher Score:**

See meta-review.

**Justification For Why Not Lower Score:**

N/A

---

### Decision · Program_Chairs · 2024-01-16

Reject